# ImpQuant: Fine-Grained Importance-Aware Quantization for Large Vision-Language Models

**Jundong Zhou** [1]  **Tianao Cai** [1]  **Yujie Huang** [4]  **Xinbing Wang** [1]  **Guang-Zhong Yang** [1]  **Nanyang Ye** [1 2 3]

## Abstract

Large Vision–Language Models (LVLMs) have demonstrated remarkable capabilities across diverse multimodal tasks, yet their high inference costs necessitate low-bit deployment. Existing post-training quantization (PTQ) pipelines primarily adopt methodologies from text-only LLMs by treating multimodal inputs as homogeneous sequences, overlooking the heterogeneous information density inherent in LVLMs. In this work, we present ImpQuant, an importance-aware PTQ framework tailored for LVLMs that mitigates low-bit accuracy degradation via fine-grained token-importance reweighted calibration and outlier-aware activation quantization. Our key insight is that quantization errors on decision-critical tokens disproportionately impact overall model behavior. Accordingly, we reweight the calibration loss using aggregated attention for textual tokens and a contextual redundancy metric for visual tokens, respectively. Across multiple LVLM backbones and diverse multimodal benchmarks, our approach consistently improves accuracy at low bitwidth and reduces quantization-induced object hallucinations compared to state-of-the-art PTQ baselines. [1]

## 1. Introduction

Large Vision–Language Models (LVLMs) have recently achieved remarkable performance on a wide range of multimodal tasks, including visual question answering (Li et al., 2023a), object detection and grounding (Peng et al., 2023),

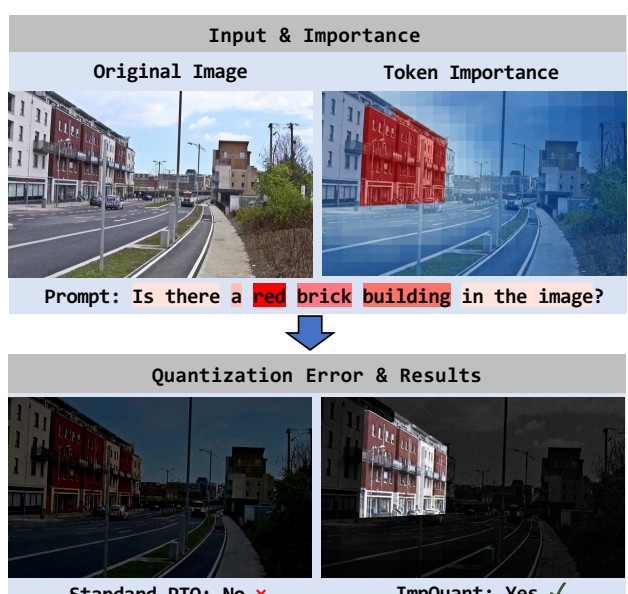

*Figure 1.* Intuition of ImpQuant. For a VQA query, we identify the importance of each textual and visual token via attention aggregation and visual redundancy, respectively. Token importance is visualized by color intensities on image patches and textual tokens, with darker red indicating greater importance. We overlay an opacity-coded distortion map to show each token's quantization error, defined as the feature distance between quantized and full-precision models. Standard PTQ induces uniform distortion across tokens under low-bit deployment, which misleads the LVLM output, whereas ImpQuant suppresses distortion on high-importance tokens, thereby preserving the correct answer.

and complex reasoning (Wang et al., 2024b). However, these models typically comprise billions of parameters and require processing high-resolution visual inputs that translate into massive token sequences, coupled with long-context Transformer decoding. These factors collectively impose substantial memory and latency burdens during inference. Post-training quantization (PTQ) has emerged as a vital strategy for practical deployment, as it reduces the memory footprint and accelerates inference without the overhead of retraining.

A prevalent approach is to directly adopt mature Large Language Model (LLM) quantization pipelines by treating the LVLM simply as a language model with appended vision modules. In this vein, widely used PTQ methods such as

[1]Shanghai Jiao Tong University, Shanghai, China [2]Shanghai Innovation Institute, Shanghai, China [3]Shanghai Artificial Intelligence Laboratory, Shanghai, China [4]Southeast University, Nanjing, China. Correspondence to: Nanyang Ye <ynylincolncam@gmail.com>.

*Proceedings of the 43rd International Conference on Machine Learning*, Seoul, South Korea. PMLR 306, 2026. Copyright 2026 by the author(s).

GPTQ (Frantar et al., 2022), AWQ (Lin et al., 2024), and SmoothQuant (Xiao et al., 2023) are directly applied to multimodal stacks, and low-bit finetuning recipes like QLoRA (Dettmers et al., 2023) are adopted where additional adaptation is permitted. However, recent studies suggest that such direct porting is suboptimal for LVLMs: under the same bitwidth, accuracy degradation tends to be more severe compared to text-only models, exhibiting high instability across tasks and prompts (Li et al., 2025; Wang et al., 2024a). This performance gap stems from multimodal-specific factors that are largely overlooked by modality-agnostic objectives. First, calibration in LLM PTQ implicitly relies on the end loss, which fails to explicitly account for the distinct information density of different modalities. In LVLMs, the massive visual context is inherently redundant, often overshadowing the sparse visual cues that are critical for reasoning. Consequently, gradient signals derived from standard objectives tend to be dominated by the abundant but low-utility visual tokens or strong language priors, leading to a quantization metric that under-penalizes errors on decision-critical visual evidence. Second, LVLM inference involves cross-modal fusion that tightly couples quantization errors across modalities and layers; minor distortions introduced in early vision-to-language alignment can be amplified downstream. Thus, indiscriminately applying text-centric objectives treats the massive visual context as homogeneous sequences, failing to capture the fine-grained structural sensitivities inherent in multimodal architectures.

Motivated by these challenges, recent studies have begun to tailor PTQ to vision–language architectures. MBQ (Li et al., 2025) introduces a modality-balanced calibration objective that accounts for the different sensitivities of vision and language tokens in LVLMs. Q-VLM (Wang et al., 2024a) proposes a PTQ framework for LVLMs that optimizes layer-wise rounding functions by modeling cross-layer dependencies for more accurate low-bit deployment. FIMA-Q (Wu et al., 2025a) and APHQ-ViT (Wu et al., 2025b) refine importance estimation and reconstruction objectives for ViT-based backbones, improving the robustness of low-bit vision transformers. While these methods enhance multimodal awareness, they are limited by their coarse granularity. Crucially, as calibration and importance modeling are still operated at the modality or block level, the reconstruction budget is misallocated to the vast number of redundant, low-impact tokens rather than the sparse, information-rich tokens that dominate cross-modal interactions and the downstream generation.

To address these limitations, our approach is grounded in the observation of visual redundancy inherent in LVLM inputs (Shang et al., 2024a; Zhang et al., 2025). In many cases, the effective visual evidence is concentrated within a small subset of visual tokens, while a large fraction of tokens contribute minimally to cross-modal interactions. Therefore,

a uniform or coarse-grained calibration objective is prone to expending the reconstruction budget on redundant positions that constitute the majority yet have low impact.

In this paper, we propose ImpQuant, an importance-aware PTQ framework that unifies weight and activation quantization for LVLMs. For weights, ImpQuant introduces a fine-grained redundancy-based token-importance metric to reweight the reconstruction objective, thereby prioritizing quantization accuracy for decision-critical tokens. For activations, leveraging the observation that outlier channels are relatively stable within each layer, we employ a two-scale quantizer that preserves outliers while maintaining high-resolution for subtle non-outlier features. ImpQuant is orthogonal to other efficiency strategies in LVLM inference such as visual token reduction (Shang et al., 2024b) and KV-cache compression for long-context decoding (Hooper et al., 2024; Liu et al., 2024c; Zhang et al., 2023), and can be integrated with them without altering their core mechanisms.

The main contributions of the paper can be summarized as:

- We propose ImpQuant, an importance-aware PTQ framework that explicitly addresses the heterogeneous token importance in LVLMs. We introduce a fine-grained token-level calibration objective that integrates a visual redundancy-based metric and attention aggregation to prioritize decision-critical tokens over redundant ones.

- We propose an outlier-aware activation quantization scheme that exploits the channel-wise stability of outliers. By employing a two-scale quantizer, we preserve high-magnitude outliers without compromising the resolution of subtle non-outlier features.

- Extensive experiments across diverse LVLM backbones and multimodal benchmarks demonstrate that ImpQuant consistently improves low-bit accuracy and mitigates object hallucinations compared with representative state-of-the-art PTQ methods.

**Conflict of Interest Disclosure.** The authors declare no financial or other substantive conflicts of interest that could reasonably be perceived to influence this work. This work was supported by New Generation Artificial Intelligence-National Science and Technology Major Project (No. 2025ZD0122901) and the National Science Foundation of China (Nos. 62572313 and 62106139). The evaluated LVLM backbones are publicly available research models, and the authors do not have financial interests in the systems or products evaluated in this paper.

## 2. Related Work

Quantization has emerged as an indispensable tool for deploying LLMs under tight memory and latency constraints, and many LVLMs simply adopt these techniques as their default compression baselines. For instance, LLM.int8() (Dettmers et al., 2022) introduces a mixed-precision decomposition scheme that specifically handles outlier channels. GPTQ (Frantar et al., 2022) proposes a one-shot, Hessian-guided layer-wise weight quantization algorithm that has become a standard PTQ baseline for autoregressive transformers. SmoothQuant (Xiao et al., 2023) mitigates activation quantization errors by migrating the difficulty to weights via per-channel rescaling. AWQ (Lin et al., 2024) is a hardware-friendly weight-only method that selectively preserves a small fraction of salient weights based on activation magnitudes. However, directly applying these text-centric methods to LVLMs fails to account for the inherent modality imbalance and visual token redundancy.

Recently, researchers shifted focus towards designing quantization schemes that explicitly target multimodal structures. LVLM-Compress-Bench (Kundu et al., 2025) provides a comprehensive empirical study of weight and KV-cache compression across multiple structures and multimodal benchmarks, revealing that different strategies exert heterogeneous effects on recognition, reasoning, and hallucination. Methodologically, Q-VLM (Wang et al., 2024a) minimizes end-to-end quantization noise by modeling the cross-layer dependency of discretization errors and optimizing layerwise rounding via activation-entropy partitioning. MBQ (Li et al., 2025) balances reconstruction loss across modalities by employing gradients of supervised calibration loss as indicators. Despite these advancements, existing methods still primarily rely on coarse-grained calibration objectives, failing to align quantization with the fine-grained structural sensitivities of long and visually redundant multimodal inputs. Unlike these approaches, ImpQuant operates at the token level, explicitly disentangling redundant visual context from decision-critical evidence.

## 3. Methodology

### 3.1. Preliminary

We consider a generic PTQ pipeline for Transformer-based LVLMs. Each block comprises Multi-Head Self-Attention (MHSA) and Feed-Forward Network (FFN) layers, where linear projections are the primary targets for quantization. For a linear projection with weight matrix $W \in \mathbb{R}^{d_{out} \times d_{in}}$ (where $d_{out}$ and $d_{in}$ denote the output and input dimensions). Let $X = [X_T; X_V] \in \mathbb{R}^{d_{in} \times N}$ denote the input activation matrix derived from a calibration set. The sequence length $N = N_t + N_v$ consists of $N_t$ textual tokens and $N_v$ visual tokens, with $X_T \in \mathbb{R}^{d_{in} \times N_t}$ and $X_V \in \mathbb{R}^{d_{in} \times N_v}$

representing the sub-matrices for text and vision, respectively.

Standard PTQ aims to find a quantized weight matrix $\widehat{W}$ that minimizes the squared reconstruction error

$$\min_{\widehat{W}} \|\widehat{W} \cdot X - W \cdot X\|_F^2 \tag{1}$$

which implicitly treats all tokens uniformly. However, given that the contributions of different tokens to the final output of an LVLM vary significantly (Ye et al., 2024), we reformulate this as a token-weighted objective:

$$\min_{\widehat{W}} \|(\widehat{W} - W) \cdot X_T \cdot P_T\|_F^2 + \|(\widehat{W} - W) \cdot X_V \cdot P_V\|_F^2,$$

where $P_T \in \mathbb{R}^{N_t \times N_t}$ and $P_V \in \mathbb{R}^{N_v \times N_v}$ are diagonal token-importance matrices that reweight the loss contribution of each textual and visual token, respectively.

### 3.2. Motivation

**Token Importance Heterogeneity.** We initiate our study with a preliminary experiment to verify token importance heterogeneity and assess the impact of incorporating such distinctions into quantization. Specifically, we compare three reweighted reconstruction calibration objectives: (i) uniform weighting, (ii) prioritizing the top 25% high-importance textual tokens, and (iii) prioritizing the top 25% high-importance visual tokens. Prioritization is implemented by applying a $10 \times$ weight multiplier to the loss contribution of the selected tokens.

For textual tokens, importance is approximated by the aggregated attention scores, which emphasizes content-bearing words (e.g., entities and attributes) over function words. Biasing the calibration objective toward these tokens produces a more informative calibration signal than uniform weighting, as shown in Table 1.

For visual tokens, we estimate importance using a redundancy probe that quantifies contextual necessity (details in Section 3.3). Table 1 reveals pronounced heterogeneity: prioritizing non-redundant visual tokens improves downstream accuracy significantly. This confirms that visual inputs contain substantial redundancy and that LVLMs often depend on sparse visual evidence for decision making.

*Table 1.* Quantization performance across different token-level reweighting strategies on Qwen2.5-VL-7B under the W3A16 setting. "10×" indicates the multiplier applied to the loss contribution of the selected tokens during calibration.

| Strategy | VizWiz (↑) | TextVQA (↑) |
|---|---|---|
| Full Precision (FP16) | 70.53 | 83.02 |
| Uniform | 67.09 | 79.03 |
| 10× Imp. Visual Tokens | 68.38 | 80.64 |
| 10× Imp. Textual Tokens | 67.75 | 79.96 |

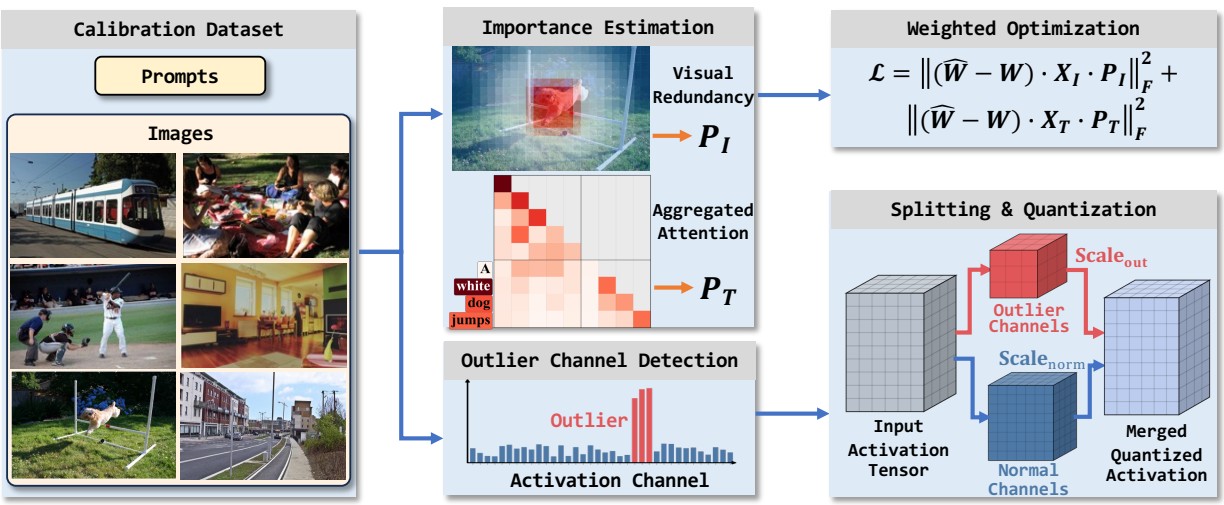

*Figure 2.* Overview of ImpQuant. Given a pretrained LVLM and a calibration set, we extract (i) token-wise importance signals for both modalities and (ii) activation statistics. Text token importance is computed by aggregating attention, while visual token importance is estimated from redundancy. These importance scores are then integrated into the calibration objective to reweight the reconstruction loss. In parallel, we summarize activation magnitudes across inputs to identify a layer-wise outlier channel set whose responses are consistently large. Activation channels are then split and scaled separately based on this outlier set to preserve precision.

**Activation Outliers.** We further inspect activation statistics by performing forward passes on multiple image-text pairs, observing a highly non-uniform pattern as depicted in the outlier channel identifier in Figure 2. Specifically, only a small subset of channels consistently exhibit disproportionately large magnitudes. Notably, these outliers demonstrate cross-input stability, appearing at consistent channel indices regardless of input content.

These input-invariant outliers make global, max-dominated scaling suboptimal under low-bit settings. A few extreme channels dictate the quantization range, thereby collapsing the effective resolution for the majority of non-outlier activations and amplifying error propagation in subsequent layers. This behavior highlights the necessity of strategies that explicitly address channel-wise heterogeneity.

Driven by these two observations, we propose ImpQuant, a PTQ framework for LVLMs that (i) modulates the calibration objective based on token importance, and (ii) incorporates an outlier-aware activation quantization scheme to decouple low-bit scaling from sparse extreme channels.

### 3.3. Importance-Aware Quantization

**Token Importance-aware weight quantization** We derive token importance scores over a calibration dataset $\mathcal{D}_{\text{cal}}$ using a hybrid strategy that differentiates between semantic density in text and redundancy in vision. Let $\mathcal{I}_t$ and $\mathcal{I}_v$ denote the sets of indices for textual and visual tokens, respectively.

For a textual token $i \in \mathcal{I}_t$, importance reflects the global attention received from the sequence. Let $A^{(h)} \in \mathbb{R}^{N \times N}$

denote the softmax attention matrix of head $h$, where $A^{(h)}_{k,i}$ represents the attention weight assigned to token $i$ by token $k$. We compute the importance weight $p_i$ as:

$$p_i = L_t \frac{1}{H} \sum_{h=1}^{H} \sum_{k=1}^{N} A^{(h)}_{k,i}, \quad i \in \mathcal{I}_t \tag{2}$$

where $L_t$ serves as the global scaling factor for textual tokens, and $H$ is the number of attention heads.

For a visual token $i \in \mathcal{I}_v$, we quantify its importance inversely to a redundancy score $r_i$, constructed via two probes. First, we assess representational dominance by repeating the token across all visual positions to form $\widetilde{X}^{(i)}$ and measuring the model's confidence in the first predicted token. Higher confidence implies that the token individually captures sufficient semantic content, corresponding to lower redundancy. Second, we assess contextual necessity via leave-one-out ablation: replacing the $i$-th visual token with a zero vector to form $X^{(-i)}$ and calculating the Jensen-Shannon Divergence (JSD) against the original distribution. A larger divergence signifies that the token is contextually indispensable, thereby indicating lower redundancy. We formulate the redundancy $r_i$ and the corresponding importance $p_i$ as:

$$r_i \triangleq -\alpha \log \max_y f_\theta(y \mid \widetilde{X}^{(i)})$$

$$+ [\text{JSD}(f_\theta(\cdot|X^{(-i)}), f_\theta(\cdot|X)) + \epsilon]^{-1}, \quad p_i = \frac{L_v}{r_i},$$

where $\alpha$ is a balancing hyperparameter, $f_\theta(\cdot)$ denotes the next-token distribution and $L_v$ is the global scaling factor for visual tokens. This fine-grained scheme supersedes identity or coarse modality-level weights by explicitly allocating the quantization budget to decision-critical tokens. Our metric

generally yields higher importance for textual tokens due to their high information density while assigning lower scores to visual tokens, aligning with observations that a substantial fraction of visual context can be compressed with minimal impact (Zhang et al., 2025). Notably, although this probing strategy necessitates $2N_V$ forward passes per image, the computational overhead is negligible in practice since PTQ calibration relies on a small, fixed dataset.

After obtaining the importance matrix $P = \text{diag}(P_T, P_V)$ as the block-diagonal concatenation of the textual and visual importance matrices, we solve for the quantized weights via a second-order objective. Let $X \in \mathbb{R}^{d_{in} \times N}$ denote the calibration activation collected on $\mathcal{D}_{\text{cal}}$ at layer $\ell$. We define the token-weighted Hessian:

$$H_\ell \triangleq X P^2 X^\top \in \mathbb{R}^{d_{in} \times d_{in}}. \quad (3)$$

We then perform per-group quantization to compress weights to a target bit-width $b_W$ and solve for the quantized weight matrix to minimize the quadratic error in a group-wise manner. Specifically, we partition the input dimension $\{1, \ldots, d_{in}\}$ into $G_\ell$ disjoint groups $\{\mathcal{G}_{\ell,g}\}_{g=1}^{G_\ell}$. Let $w_{u,g} \in \mathbb{R}^{|\mathcal{G}_{\ell,g}|}$ denote the sub-vector of the $u$-th row of $W$ restricted to indices in $\mathcal{G}_{\ell,g}$. Similarly, $H_{g,g}$ denotes the principal sub-matrix of $H_\ell$ indexed by $\mathcal{G}_{\ell,g}$, and $H_{g,>g}$ denotes the cross-block matrix between $\mathcal{G}_{\ell,g}$ and remaining unquantized indices.

We process the groups sequentially from $g = 1$ to $G_\ell$. At step $g$, we solve the local subproblem

$$\hat{w}_{u,g} = \arg \min_{q \in \mathcal{Q}_{b_W}^{|\mathcal{G}_{\ell,g}|}} (w_{u,g} - q)^\top H_{g,g} (w_{u,g} - q), \quad (4)$$

where $\mathcal{Q}_{b_W}^{|\mathcal{G}_{\ell,g}|}$ denotes the set of available $b_W$-bit quantized weights determined via min-max clipping and rounding. Defining the quantization error $e_{u,g} \triangleq w_{u,g} - \hat{w}_{u,g}$, we compensate for the remaining unquantized weights using the second-order coupling in $H_\ell$:

$$w_{u,>g} \leftarrow w_{u,>g} - e_{u,g} H_{g,g}^{-1} H_{g,>g} \quad (5)$$

After iterating through all groups, the aggregated quantized groups form the final quantized weight matrix $\widehat{W}$.

**Channel Outlier Aware Activation Quantization**  In per-token activation quantization, sparse outlier channels can dominate the dynamic range, inflating the step size and eroding the representation of subtle features. For a fixed layer $\ell$, such outlier channels are spatially consistent across different inputs. Therefore, we identify a layer-wise outlier channel set $J_\ell \subset \{1, \ldots, d_{in}\}$ based on average activation magnitudes computed during forward passes over $\mathcal{D}_{cal}$. We then quantize outlier and non-outlier channels using separate scales.

Let $x_{\ell,i} \in \mathbb{R}^{d_{in}}$ be the activation of token $i$ at layer $\ell$. We decompose it into outlier $x_{\ell,i}^H$ and normal $x_{\ell,i}^L$ components by masking:

$$x_{\ell,i}^H = x_{\ell,i} \odot \mathbf{1}_{\mathcal{J}_\ell}, \quad x_{\ell,i}^L = x_{\ell,i} \odot (1 - \mathbf{1}_{\mathcal{J}_\ell}), \quad (6)$$

where $\odot$ denotes element-wise multiplication and $\mathbf{1}_{\mathcal{J}_\ell}$ is the binary mask for indices in $\mathcal{J}_\ell$. We compute separate step sizes for each component based on the target activation bit-width $b_A$:

$$\Delta_{\ell,i}^H = \frac{\max_{j \in J_\ell} \left| (x_{\ell,i})_j \right|}{2^{b_A-1} - 1}, \quad \Delta_{\ell,i}^L = \frac{\max_{j \notin J_\ell} \left| (x_{\ell,i})_j \right|}{2^{b_A-1} - 1},$$

Quantization is applied separately:

$$\widehat{x}_{\ell,i}^S = \Delta_{\ell,i}^S \cdot \text{clip}\left( \text{round}\left( \frac{x_{\ell,i}^S}{\Delta_{\ell,i}^S} \right), -2^{b_A-1}, 2^{b_A-1} - 1 \right),$$

for each group $S \in \{H, L\}$. The final quantized activation is reconstructed by summing the components:

$$\widehat{x}_{\ell,i} = \widehat{x}_{\ell,i}^H + \widehat{x}_{\ell,i}^L. \quad (7)$$

This design preserves the information carried by outlier channels via an outlier-specific scale, while maintaining fine-grained resolution for the majority of non-outlier channels.

### 3.4. Theoretical Analysis

In this section, we provide a theoretical justification for reweighting quantization errors based on token redundancy. We analyze the sensitivity of the PTQ-induced loss change, demonstrating that redundant tokens induce spectral degeneracy in the Hessian matrix. Consequently, the loss becomes locally insensitive to quantization errors along redundant directions.

Let $h = \text{vec}([h_1; \ldots; h_N]) \in \mathbb{R}^{N d_{in}}$ denote the vectorized concatenated token representations. We consider the Hessian of the task loss $\mathcal{L}$ with respect to these activations, $H \triangleq \mathbb{E}[\nabla_h^2 \mathcal{L}] \in \mathbb{R}^{N d_{in} \times N d_{in}}$. To quantify redundancy mathematically, we consider the linear predictability of the $i$-th token given its context $h_{-i}$. We define the conditional covariance matrix via the Schur complement of the joint covariance $\Sigma^{(h)} = \text{Cov}(h)$:

$$\Sigma_{i|-i}^{(h)} \triangleq \Sigma_{ii}^{(h)} - \Sigma_{i,-i}^{(h)} \left( \Sigma_{-i,-i}^{(h)} \right)^{-1} \Sigma_{-i,i}^{(h)}, \quad (8)$$

where $\Sigma_{i|-i}^{(h)}$ is the conditional covariance of token $i$ given the context $h_{-i}$. We quantify the non-redundant information by the trace of the conditional covariance matrix.

**Proposition 3.1.** (*Redundancy induces spectral degeneracy*) *Assume the model is well-trained such that it does not rely on noise orthogonal to the signal structure for prediction (Efficient Coding Hypothesis). Let $v$ be any direction in the activation space primarily aligned with the residual*

*Table 2.* Comparative performance of quantized LVLMs on multiple VQA benchmarks. Results are reported in accuracy (%).

| Model | Bitwidth | Method | VizWiz | ScienceQA | MMBench | TextVQA | OCRBench | MMMU | Average (↑) |
|---|---|---|---|---|---|---|---|---|---|
| LLaVA-OneVision-7B | FP16 | - | 60.41 | 89.98 | 80.84 | 75.98 | 62.30 | 49.22 | 69.79 |
| | W3A16 | RTN | 58.63 | 88.02 | 77.74 | 63.56 | 48.40 | 43.35 | 63.28 |
| | | AWQ | 59.23 | 88.24 | 78.52 | 66.54 | 53.10 | 45.31 | 65.16 |
| | | MBQ | 59.34 | 88.46 | 78.09 | 66.62 | 52.40 | 44.57 | 64.91 |
| | | Q-VLM | 58.04 | 88.52 | 78.06 | 65.57 | 52.40 | 44.84 | 64.57 |
| | | SmoothQuant | 58.13 | 87.76 | 78.23 | 65.49 | 39.30 | 43.92 | 62.14 |
| | | ImpQuant | **59.93** | **88.69** | **79.69** | **70.13** | **57.20** | **45.82** | **66.91** |
| | W4A8 | RTN | 58.10 | 87.86 | 77.84 | 61.90 | 39.40 | 42.89 | 61.33 |
| | | MBQ | 59.00 | 88.12 | 77.23 | 68.41 | 53.20 | **45.78** | 65.29 |
| | | Q-VLM | 57.84 | 88.25 | 77.46 | 65.31 | 49.80 | 43.77 | 63.41 |
| | | SmoothQuant | 56.46 | 87.67 | 77.84 | 57.44 | 30.80 | 40.33 | 58.42 |
| | | ImpQuant | **59.44** | **88.46** | **78.85** | **70.66** | **56.80** | 45.60 | **66.64** |
| Qwen2.5-VL-7B | FP16 | - | 70.53 | 88.45 | 82.82 | 83.02 | 83.90 | 50.95 | 76.61 |
| | W3A16 | RTN | 67.09 | 85.31 | 79.98 | 79.03 | 75.70 | 45.91 | 72.17 |
| | | AWQ | 67.83 | 84.51 | 80.42 | 80.36 | 79.30 | 43.14 | 72.59 |
| | | MBQ | 65.77 | 85.59 | 80.33 | 80.80 | 76.50 | **46.86** | 72.64 |
| | | Q-VLM | 65.82 | 85.29 | 79.23 | 79.66 | 77.20 | 45.41 | 72.10 |
| | | SmoothQuant | 66.23 | 85.19 | 79.38 | 79.69 | 72.70 | 42.86 | 71.01 |
| | | ImpQuant | **68.77** | **86.32** | **81.36** | **81.33** | **82.00** | 46.48 | **74.38** |
| | W4A8 | RTN | 65.94 | 85.24 | 78.35 | 77.23 | 65.20 | **46.67** | 69.77 |
| | | MBQ | 65.86 | 85.26 | 78.52 | 75.41 | 69.30 | 45.24 | 69.93 |
| | | Q-VLM | 65.48 | 85.52 | 78.06 | 76.86 | 73.30 | 45.43 | 70.78 |
| | | SmoothQuant | 64.87 | 83.26 | 77.45 | 77.37 | 72.20 | 43.43 | 69.76 |
| | | ImpQuant | **67.51** | **86.02** | **79.74** | **80.88** | **82.90** | 46.04 | **73.85** |
| InternVL2-8B | FP16 | - | 60.94 | 96.25 | 81.96 | 77.02 | 76.70 | 48.29 | 73.53 |
| | W3A16 | RTN | 56.07 | 95.68 | 77.32 | 74.54 | 74.00 | 43.14 | 70.13 |
| | | AWQ | 58.61 | 95.59 | 79.04 | 75.09 | 75.50 | 44.48 | 71.39 |
| | | MBQ | **59.09** | 95.78 | 79.04 | 74.68 | 74.90 | 44.86 | 71.39 |
| | | Q-VLM | 58.03 | 95.64 | 78.92 | 75.12 | 75.30 | 44.25 | 71.21 |
| | | SmoothQuant | 58.46 | 95.61 | 78.87 | 74.43 | 75.70 | 44.95 | 71.34 |
| | | ImpQuant | 58.99 | **95.88** | **79.44** | **75.46** | **76.10** | **45.33** | **71.87** |
| | W4A8 | RTN | 56.90 | 95.35 | 77.15 | 73.09 | 73.20 | 44.29 | 70.00 |
| | | MBQ | 56.25 | 94.97 | **78.44** | 72.34 | 73.10 | 43.52 | 69.77 |
| | | Q-VLM | 54.93 | 95.28 | 77.86 | 72.26 | 73.80 | 43.86 | 69.67 |
| | | SmoothQuant | 52.09 | 95.05 | 76.03 | 72.45 | 71.90 | 43.43 | 68.49 |
| | | ImpQuant | **59.73** | **95.40** | 78.35 | **75.48** | 75.10 | **45.27** | **71.56** |

error of a highly redundant token $i$ (i.e. aligned with the eigenvectors of $\Sigma_{i|-i}^{(h)}$). Then, the curvature of the loss along this direction is bounded by the redundancy:

$$v^\top H v \le C\,\xi_i, \quad \xi_i \triangleq \mathrm{Tr}\left(\Sigma_{i|-i}^{(h)}\right), \qquad (9)$$

for some constant $C > 0$.

*Remark* 3.2. Detailed proof is provided in Appendix B. Although we analyze activation sensitivity, weight quantization error $\Delta W$ translates directly into activation perturbation $\Delta h = \Delta W \cdot x$. Proposition 3.1 establishes that highly redundant tokens (small $\xi_i$) induce near-zero curvature directions in the loss landscape. Quantization errors aligned with these directions entail negligible loss penalties, even if their magnitude is large. This justifies our objective of assigning smaller calibration weights to redundant tokens to

prevent the optimization from overfitting to these insensitive directions, while emphasizing non-redundant tokens to align calibration with decision-critical evidence.

## 4. Experiments

### 4.1. Experimental Settings

**Baselines** We benchmark ImpQuant against the FP16 baseline and representative PTQ methods. For the W3A16 (weight-only) setting, baselines include RTN, AWQ, MBQ, Q-VLM, and SmoothQuant, where activations are kept in full precision. For the W4A8 setting, we compare against RTN, MBQ, Q-VLM, and SmoothQuant. Note that AWQ is excluded from the W4A8 comparison as it is specifically designed as a weight-only quantizer.

*Table 3.* Evaluation of object hallucination of quantized LVLMs on the POPE benchmark. We report F1-score and Accuracy (%).

| Model | Bitwidth | Method | Popular | | Random | | Adversarial | | Average | |
|---|---|---|---|---|---|---|---|---|---|---|
| | | | F1-score | Acc | F1-score | Acc | F1-score | Acc | F1-score (↑) | Acc (↑) |
| Qwen2.5-VL-7B | FP16 | - | 86.44 | 87.70 | 87.34 | 88.63 | 85.51 | 86.70 | 86.43 | 87.68 |
| | W3A16 | RTN | 83.58 | 85.67 | 83.96 | 86.07 | 83.08 | 85.13 | 83.54 | 85.62 |
| | | AWQ | 84.72 | 86.43 | 85.45 | 87.20 | 83.71 | 85.37 | 84.63 | 86.33 |
| | | MBQ | 85.73 | 86.47 | 86.20 | 87.77 | 84.62 | 86.10 | 85.52 | 86.78 |
| | | Q-VLM | 85.02 | 86.60 | 85.92 | 87.53 | 84.05 | 85.57 | 85.00 | 86.57 |
| | | SmoothQuant | **86.62** | 87.33 | **87.66** | 88.43 | 85.31 | 85.77 | **86.53** | 87.18 |
| | | ImpQuant | 86.45 | **88.17** | 87.07 | **88.82** | **85.92** | **86.53** | 86.48 | **87.84** |
| | W4A8 | RTN | 84.24 | 86.07 | 85.01 | 86.87 | 83.73 | 85.53 | 84.33 | 86.16 |
| | | MBQ | **84.70** | 84.37 | 85.50 | **87.20** | **83.80** | 85.43 | **84.67** | 85.67 |
| | | Q-VLM | 83.58 | 84.02 | 85.06 | 86.26 | 83.50 | 84.88 | 84.05 | 85.05 |
| | | SmoothQuant | 82.48 | 84.73 | 83.34 | 85.63 | 82.21 | 84.43 | 82.68 | 84.93 |
| | | ImpQuant | 84.34 | **86.10** | **85.54** | 87.03 | 83.77 | **85.50** | 84.55 | **86.21** |
| InternVL2-8B | FP16 | - | 86.72 | 87.80 | 87.80 | 88.93 | 85.82 | 86.80 | 86.78 | 87.84 |
| | W3A16 | RTN | 86.62 | 87.63 | 87.76 | 88.83 | 85.06 | 85.93 | 86.48 | 87.46 |
| | | AWQ | 87.63 | 88.43 | 88.61 | 89.47 | **85.83** | **86.50** | 87.36 | 88.13 |
| | | MBQ | 87.35 | 88.27 | 88.11 | 89.07 | 85.39 | 86.13 | 86.95 | 87.82 |
| | | Q-VLM | 86.79 | 87.70 | 88.05 | 89.03 | 85.11 | 85.87 | 86.65 | 87.53 |
| | | SmoothQuant | **87.95** | 87.33 | **88.92** | 88.60 | 85.39 | 84.53 | 87.42 | 86.82 |
| | | ImpQuant | 87.66 | **88.48** | 88.82 | **89.71** | 85.75 | 86.36 | **87.41** | **88.18** |
| | W4A8 | RTN | 82.65 | 84.80 | 83.13 | 85.30 | 82.02 | 84.10 | 82.60 | 84.73 |
| | | MBQ | 83.34 | 84.73 | 83.92 | 85.47 | 82.85 | 84.27 | 83.37 | 84.82 |
| | | Q-VLM | 84.26 | 84.88 | 85.12 | 85.74 | 83.48 | 84.50 | 84.29 | 85.04 |
| | | SmoothQuant | 87.03 | 87.97 | 87.75 | 88.73 | 85.28 | 86.07 | 86.69 | 87.59 |
| | | ImpQuant | **87.34** | **88.30** | **88.23** | **89.23** | **86.00** | **86.83** | **87.19** | **88.12** |

**Models**   We conduct experiments on three representative open-source LVLM backbones: LLaVA-OneVision-7B, Qwen2.5-VL-7B and InternVL2-8B, covering diverse architectural families and vision encoders.

**Datasets**   We evaluate the quantized model's performance across a comprehensive suite of Visual Question-Answering (VQA) benchmarks. These include VizWiz (Gurari et al., 2018) and MMBench (Liu et al., 2024a) for general visual comprehension; ScienceQA (Saikh et al., 2022) and MMMU (Yue et al., 2024) for discipline-level reasoning; and TextVQA (Singh et al., 2019) and OCRBench (Liu et al., 2024b) for text-centric perception. Additionally, we use POPE (Li et al., 2023b) to quantify object-level hallucinations under quantization.

**Calibration**   We enforce a unified calibration protocol: all methods and backbones utilize 256 randomly sampled images from MSCOCO (Lin et al., 2015) and the corresponding captioning text as the calibration dataset.

### 4.2. Results and Analysis

Table 2 demonstrates that ImpQuant achieves consistent improvements over PTQ baselines across multiple LVLM backbones and bit-width settings on various VQA tasks. This indicates that ImpQuant is a robust and versatile quanti-

zation method that can be seamlessly integrated into diverse architectures. The performance gains validate our hypothesis that multimodal quantization errors are structurally heterogeneous across long multimodal sequences. Since predictions are often driven by a sparse subset of decision-relevant tokens, uniform calibration misallocates the reconstruction budget to redundant features while leaving large errors on the critical tokens. By biasing the optimization towards high-importance tokens, ImpQuant suppresses quantization noise where it matters most, aligning the low-bit model with the decision-critical evidence.

A notable pattern in Table 2 is that the improvements are particularly pronounced on text-centric perception benchmarks such as TextVQA and OCRBench. These tasks demand precise alignment with sparse, spatially localized visual tokens (e.g., character regions) that possess high information density compared to the background. Baseline methods, which treat the vast background and small character regions equally, fail to preserve these fine-grained details. In contrast, ImpQuant explicitly up-weights these non-redundant tokens, ensuring that sharp visual cues required for text recognition are preserved amidst the compression noise. Consequently, importance-aware calibration yields more substantial performance benefits in these tasks.

For reliability, Table 3 reports the POPE results, where

*Table 4.* Ablation study analyzing the impact of each component for Qwen2.5-VL-7B under the W4A8 setting. "W-imp" denotes token importance-aware weight quantization, and "A-out" denotes channel outlier-aware activation quantization.

| Bitwidth | Model | VizWiz | ScienceQA | MMBench | TextVQA | OCRBench | MMMU | Average (↑) |
|---|---|---|---|---|---|---|---|---|
| | ImpQuant | 67.51 | 86.02 | 79.74 | 80.88 | 82.90 | 46.04 | 73.85 |
| W4A8 | ImpQuant w.o. W-imp | 65.92 | 85.44 | 79.01 | 77.89 | 74.70 | 45.31 | 71.38 |
| | ImpQuant w.o. A-out | 67.14 | 85.39 | 79.24 | 80.43 | 82.30 | 45.50 | 73.33 |

object-level hallucination is assessed via F1-score and accuracy across popular, random, and adversarial splits. Additional results for LLaVA-OneVision-7B are provided in Appendix C. Interestingly, the results suggest that quantization does not monotonically exacerbate hallucination. A plausible explanation is that quantization noise can potentially act as a mild regularizer to mitigate overconfidence, rendering the model less prone to "default yes" responses in ambiguous cases (Kundu et al., 2025). Beyond this effect, ImpQuant provides a more robust improvement attributed to the following two factors. First, importance-aware calibration preferentially preserves the token-conditioned computations that carry cross-modal evidence; therefore, the model is less likely to lose the key visual cues required to refute an incorrect object claim. Second, our outlier-aware activation scaling mitigates maximum-dominated quantization ranges caused by sparse high-magnitude channels, preventing resolution collapse in non-outlier channels. Together, these mechanisms help maintain high-fidelity cross-modal alignment signals under low-bit inference, reducing hallucinations more reliably than baselines.

### 4.3. Ablation Study

The ablation results in Table 4 decouple the effects of token-importance reweighting in weight calibration and outlier-aware activation scaling under the W4A8 setting on Qwen2.5-VL-7B.

**Impact of Token-Importance Reweighting.** We replace the token importance matrix $P$ with the identity matrix $I$, thereby assigning uniform loss weights to all tokens. This simplification results in a consistent performance degradation across all tasks, especially for text-heavy benchmarks like OCRBench and TextVQA, where token importance plays a critical role in guiding the quantization process towards decision-critical tokens. This trend is consistent with our main results, where importance-aware calibration provides larger gains in text-centric scenarios.

Under uniform weighting, the calibration objective is statistically dominated by the vast number of frequent but low-impact tokens. To minimize the average loss, the optimization naturally prioritizes fitting these non-essential regions as they constitute the majority. In other words, the calibration misspends its limited degrees of freedom on fitting abundant but redundant tokens, while under-penalizing mis-

| BS | Method | Precision | TPOT | Speedup |
|---|---|---|---|---|
| | Baseline | FP16 | 45.2 | - |
| 1 | RTN | W4A8 | 30.7 | 1.47 × |
| | ImpQuant | W4A8 | 31.1 | 1.45 × |
| | Baseline | FP16 | 62.4 | - |
| 4 | RTN | W4A8 | 42.1 | 1.48 × |
| | ImpQuant | W4A8 | 42.4 | 1.47 × |

*Table 5.* Inference latency analysis measuring Time per Output Token (TPOT) in milliseconds during generation decoding on the Qwen2.5-VL-7B model. "BS" denotes batch size.

matches on the few decision-critical tokens. Consequently, after quantization, substantial errors persist on those essential tokens. This effect is more pronounced on text-centric tasks where prediction quality depends on a small set of highly informative tokens. This degradation underscores the necessity of importance-aware calibration in handling the skewed information density of LVLMs.

**Impact of Outlier-Aware Activation Quantization.** We replace our outlier-aware activation quantization scheme with standard per-token activation quantization, which employs a single token-wise scale to quantize all channels uniformly. While the performance drop is less severe than in the first case, a noticeable reduction in accuracy persists. Without outlier isolation, the quantization grid is determined by a few extreme outlier channels, leaving the vast majority of normal channels representing subtle features quantized with insufficient resolution. This ablation suggests that using a single token-wise scale is insufficient when activation magnitudes vary substantially across channels.

### 4.4. Efficiency Analysis

To verify the practical deployment benefits, we measured the per-token decoding latency of the Qwen2.5-VL-7B model on an NVIDIA A100 GPU. As shown in Table 5, ImpQuant achieves a 1.45× speedup compared to the FP16 baseline in the W4A8 setting. Regarding the algorithmic overhead introduced by our outlier-aware activation quantization, we note that while the activation is decomposed into outlier and normal components, the sum of non-zero elements across both components remains identical to the original input size. Consequently, the total number of Multiply-Accumulate (MAC) operations required for matrix multiplication which is the most compute-intensive part of inference remains invariant. The additional element-wise masking and sum-

mation operations are computationally lightweight. Empirically, ImpQuant incurs negligible latency overhead ($< 2\%$) compared to standard per-token quantization methods (e.g. RTN) while outperforming them in accuracy.

## 5. Conclusion

We propose ImpQuant, an importance-aware PTQ framework specifically tailored for LVLMs. Recognizing the heterogeneity of multimodal information, ImpQuant re-weights the calibration objective to prioritize decision-critical tokens, ensuring high-fidelity reconstruction where it matters most. In addition, our outlier-aware activation quantization effectively disentangles extreme outliers from normal channels, preventing resolution collapse under heavy-tailed activation distributions. Extensive experiments across multiple LVLM backbones and diverse VQA benchmarks demonstrate that our method consistently outperforms state-of-the-art PTQ baselines. These results establish ImpQuant as a robust and effective solution for achieving accurate and reliable low-bit LVLM deployment in practical scenarios.

## Acknowledgements

This work is supported by New Generation Artificial Intelligence-National Science and Technology Major Project (No. 2025ZD0122901). This work is also supported by National Science Foundation of China (No. 62572313, No. 62106139).

## Impact Statement

This paper presents work whose goal is to advance the field of Machine Learning. There are many potential societal consequences of our work, none which we feel must be specifically highlighted here.

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

# A. Implementation Details

Unless otherwise specified, we set $\alpha = 1.5$, $L_t = 1.0$, $L_v = 0.1$, use a weight quantization group size of 128, and select the top $1\%$ channels by average activation magnitude as the outlier set in each layer.

# B. Proof of Proposition 3.1

In this section, we provide a complete derivation underlying Proposition 3.1. The proof establishes a connection between the statistical redundancy of token representations and the local geometry of the loss landscape.

**Step 1: The Fisher Information and Spectral Alignment** Let $\mathcal{H} \triangleq \mathbb{E}[\nabla_h^2 \mathcal{L}]$ denote the Hessian of the loss with respect to the activation vector $h$. We rely on the standard approximation of the Hessian by the Empirical Fisher Information Matrix:

$$\mathcal{H} \approx \mathbb{E}\left[(\nabla_h \mathcal{L})(\nabla_h \mathcal{L})^\top\right]. \tag{10}$$

The *Efficient Coding Hypothesis* in representation learning posits that a well-trained model maximizes the mutual information between representations and the task target while minimizing the complexity. Consequently, the model's sensitivity (curvature) aligns with the data manifold: the gradient magnitude is significant primarily along directions where the data exhibits meaningful variation. Conversely, in directions orthogonal to the data manifold or where data variance is negligible (noise), the model learns to be insensitive (flat loss landscape).

Mathematically, this implies that the eigenvectors of the Hessian $\mathcal{H}$ align with the eigenvectors of the data covariance matrix $\Sigma = \mathrm{Cov}(h)$, and the eigenvalues satisfy:

$$\lambda_k(\mathcal{H}) \leq C \cdot \lambda_k(\Sigma), \tag{11}$$

for some scaling constant $C > 0$. This relationship suggests that quantization errors aligned with low-variance directions in $\Sigma$ will induce minimal increase in the loss $\mathcal{L}$.

**Step 2: Geometry Characterization of Redundancy** We formally define redundancy using the conditional covariance derived from the Schur complement. Partition the token vector $h$ into the $i$-th token $h_i$ and the context $h_{-i}$. The joint covariance is $\Sigma = \begin{pmatrix} \Sigma_{ii} & \Sigma_{i,-i} \\ \Sigma_{-i,i} & \Sigma_{-i,-i} \end{pmatrix}$.

The *residual* of $h_i$ effectively not predicted by the linear context is defined as $u_i = h_i - \Sigma_{i,-i}\Sigma_{-i,-i}^{-1}h_{-i}$. The covariance of this residual is given by the Schur complement:

$$\Sigma_{i|-i} \triangleq \mathrm{Cov}(u_i) = \Sigma_{ii} - \Sigma_{i,-i}\Sigma_{-i,-i}^{-1}\Sigma_{-i,i}. \tag{12}$$

We define the residual subspace $\mathcal{S}_\perp^{(i)}$ as the subspace spanned by the eigenvectors of $\Sigma_{i|-i}$ corresponding to non-zero eigenvalues. This subspace represents the directions of new information provided by token $i$ that cannot be inferred from $h_{-i}$. The total \*\*redundancy energy\*\* is quantified by the trace:

$$\xi_i \triangleq \mathrm{Tr}(\Sigma_{i|-i}) = \mathbb{E}[\|u_i\|_2^2]. \tag{13}$$

A smaller $\xi_i$ implies that $h_i$ is highly redundant.

**Step 3: Bounding the Curvature (Proof of Proposition 3.1)** Let $v \in \mathcal{S}_\perp^{(i)}$ be a unit vector ($\|v\|_2 = 1$) aligned with the residual error of token $i$. Since $v$ lies in the residual subspace of token $i$, its variance under the data distribution is bounded by the total residual energy:

$$v^\top \Sigma v = \mathrm{Var}(v^\top h) \leq \mathrm{Tr}(\Sigma_{i|-i}) = \xi_i. \tag{14}$$

Invoking the alignment assumption from Eq. (11), the curvature of the loss along direction $v$ is bounded by the data variance:

$$v^\top \mathcal{H} v \leq C \cdot (v^\top \Sigma v) \leq C \cdot \xi_i. \tag{15}$$

This inequality directly links the information-theoretic redundancy $\xi_i$ to the loss geometry. It demonstrates that if a token is highly redundant ($\xi_i \to 0$), the loss function becomes locally flat along the directions of quantization error for that token. This theoretically justifies our strategy of assigning lower calibration weights to redundant visual tokens, as errors on these tokens incur a negligible penalty in the task loss.

## C. More Experimental Results

To further examine the role of outlier-aware activation quantization, we evaluate a more aggressive W4A6 setting, where activation quantization errors become more pronounced. As shown in Table 6, removing A-out causes larger drops than in

*Table 6.* Ablation of outlier-aware activation quantization under a more aggressive W4A6 setting on Qwen2.5-VL-7B.

| Method | ScienceQA | MMBench | OCRBench | MMMU |
|---|---|---|---|---|
| ImpQuant | 81.38 | 77.83 | 74.80 | 42.24 |
| ImpQuant w.o. A-out | 80.71 | 77.15 | 73.60 | 41.67 |

the W4A8 setting, indicating that separating stable outlier channels becomes increasingly important as activation precision decreases.

We further evaluate whether ImpQuant is sensitive to the size or domain of the calibration set. Table 7 shows that performance is stable once the calibration set reaches a moderate size, and replacing MSCOCO with LLaVA-Instruct introduces only small changes, suggesting that the estimated importance patterns are not tied to a specific calibration subset or domain.

*Table 7.* Sensitivity to calibration set size and domain on Qwen2.5-VL-7B under W3A16.

| Calibration Setting | Size | ScienceQA | MMBench | OCRBench | MMMU |
|---|---|---|---|---|---|
| MSCOCO | 64 | 85.07 | 80.68 | 80.40 | 45.57 |
| MSCOCO | 128 | 85.94 | 81.01 | 81.60 | 46.24 |
| MSCOCO | 200 | 86.23 | 81.25 | 81.80 | 46.44 |
| MSCOCO | 256 | 86.32 | 81.36 | 82.00 | 46.48 |
| MSCOCO | 512 | 86.53 | 81.42 | 82.20 | 46.56 |
| LLaVA-Instruct | 256 | 86.21 | 81.38 | 82.20 | 46.32 |

To validate the assumption behind A-out, we measure whether the top outlier channels remain stable across calibration inputs and across different LVLM architectures. The consistently high Jaccard similarities in Table 8 indicate that the outlier channel sets are highly stable with respect to input, including on the cross-attention-based mPLUG-Owl3-7B model, supporting the use of a fixed layer-wise outlier mask.

*Table 8.* Cross-input stability of top-1% outlier channel sets measured by pairwise Jaccard similarity over 256 MSCOCO images.

| Model | Fusion Type | Avg Jaccard | Min Jaccard | Max Jaccard |
|---|---|---|---|---|
| LLaVA-OneVision-7B | MLP projector | 0.953 | 0.928 (Layer 4) | 0.989 (Layer 22) |
| Qwen2.5-VL-7B | Dynamic ViT + LLM | 0.947 | 0.922 (Layer 2) | 0.986 (Layer 20) |
| InternVL2-8B | InternViT + MLP | 0.958 | 0.931 (Layer 3) | 0.987 (Layer 24) |
| mPLUG-Owl3-7B | Cross-attention | 0.941 | 0.920 (Layer 2) | 0.985 (Layer 22) |

We test a fusion-heavy architecture to examine whether ImpQuant's gains are specific to concatenation-style LVLMs. Table 9 shows that ImpQuant remains competitive on mPLUG-Owl3-7B under both W3A16 and W4A8, suggesting that the heterogeneous-density observation extends beyond LLaVA-style architectures.

*Table 9.* Generalization to mPLUG-Owl3-7B, a cross-attention-based fusion architecture.

| Bitwidth | Method | ScienceQA | MMBench | OCRBench | MMMU |
|---|---|---|---|---|---|
| FP16 | - | 89.54 | 79.60 | 72.20 | 49.56 |
| W3A16 | RTN | 85.90 | 76.22 | 65.20 | 46.68 |
| W3A16 | AWQ | 87.25 | 77.42 | 67.80 | 46.78 |
| W3A16 | MBQ | 86.82 | 76.83 | 67.60 | 46.42 |
| W3A16 | ImpQuant | 87.86 | 77.45 | 70.80 | 48.12 |
| W4A8 | RTN | 85.44 | 76.08 | 65.30 | 45.19 |
| W4A8 | MBQ | 86.89 | 76.93 | 67.20 | 46.70 |
| W4A8 | ImpQuant | 87.52 | 77.59 | 70.30 | 48.24 |

*Table 10.* Performance of quantized LVLMs on POPE benchmark on LLaVA-OneVision-7B model.

| Model | Bitwidth | Method | Popular | | Random | | Adversarial | | Average | |
|---|---|---|---|---|---|---|---|---|---|---|
| | | | F1-score | Acc | F1-score | Acc | F1-score | Acc | F1-score | Acc |
| LLaVA-OneVision-7B | FP16 | - | 88.40 | 89.20 | 89.68 | 90.53 | 87.16 | 87.87 | 88.41 | 89.20 |
| | W3A16 | RTN | 88.71 | 89.47 | 89.84 | 90.63 | **87.60** | **88.27** | 88.72 | 89.46 |
| | | AWQ | 89.01 | 89.70 | 90.16 | 90.90 | 87.25 | 87.80 | 88.81 | 89.47 |
| | | MBQ | 89.11 | 89.73 | 90.52 | 91.20 | 87.33 | 87.83 | 88.99 | 89.59 |
| | | Q-VLM | 86.88 | 88.20 | 88.65 | 90.07 | 85.21 | 86.33 | 86.91 | 88.20 |
| | | SmoothQuant | 89.01 | 89.63 | 90.51 | 91.20 | 87.48 | 88.00 | 89.00 | 89.61 |
| | | ImpQuant | **89.83** | **89.80** | **93.05** | **93.27** | 86.78 | 86.27 | **89.89** | **90.99** |
| | W4A8 | RTN | 87.61 | 88.70 | 88.28 | 89.40 | 86.54 | 87.53 | 87.48 | 88.54 |
| | | MBQ | 87.44 | 88.50 | 88.47 | 89.57 | 86.32 | 87.30 | 87.41 | 88.46 |
| | | Q-VLM | 86.21 | 87.53 | 87.56 | 88.91 | 86.10 | 86.93 | 86.62 | 87.79 |
| | | SmoothQuant | **89.42** | 89.70 | **92.04** | **92.47** | 87.36 | 87.40 | **89.61** | 89.86 |
| | | ImpQuant | 89.03 | **89.90** | 90.13 | 91.03 | **87.81** | **88.72** | 88.99 | **89.88** |

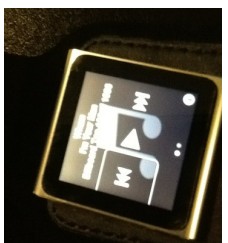

Question: What is this?

Full Precision: Watch
AWQ: Unanswerable
MBQ: Unanswerable
SmoothQuant: Unanswerable
ImpQuant: Watch

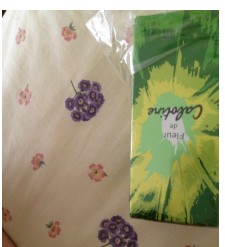

Question: What is the name of the perfume?

Full Precision: fleur de calotine
AWQ: Unanswerable
MBQ: Unanswerable
SmoothQuant: Unanswerable
ImpQuant: fleur de calotine

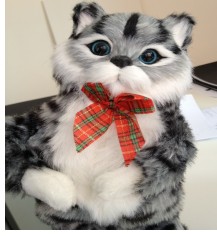

Question: What's the color of the animal?

FP: Gray and white
AWQ: White
MBQ: White
SmoothQuant: White
ImpQuant: Gray and white

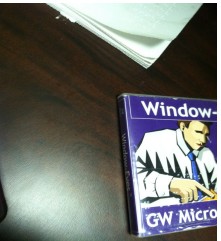

Question: What does this say?

Full Precision: Window-eyes GW Micro
AWQ: Window
MBQ: Window
SmoothQuant: Window
ImpQuant: Window eyes GW Micro

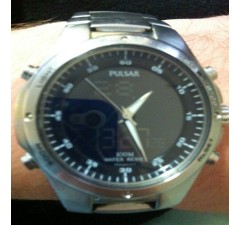

Question: What time does the watch say?

FP: 1:27
AWQ: Unanswerable
MBQ: Unanswerable
SmoothQuant: Unanswerable
ImpQuant: 1:27

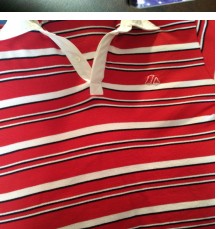

Question: Could you describe the colors of this polo shirt to me please.

Full Precision: Red, black and white
AWQ: Red
MBQ: Red
SmoothQuant: Red
ImpQuant: Red, black and white

*Figure 3.* Qualitative comparison of quantization robustness. Baseline PTQ methods often degrade key visual evidence, leading to incomplete or "Unanswerable" responses. In contrast, ImpQuant preserves high-importance tokens, enabling the model to correctly identify fine-grained visual details and generates full-precision-level answers across diverse scenarios.

