# OpenReview forum: "ImpQuant: Fine-Grained Importance-Aware Quantization for Large Vision-Language Models"
_ICML.cc/2026/Conference — ICML 2026 regular_

### Official Review · Reviewer_Axuk · 2026-03-12

**Soundness:** 3
**Presentation:** 3
**Significance:** 3
**Originality:** 3
**Overall Recommendation:** 4
**Confidence:** 3

**Summary:**

This paper proposes ImpQuant, a post-training quantization (PTQ) framework specifically designed for Large Vision-Language Models (LVLMs). The key insight is that existing PTQ methods treat multimodal inputs as homogeneous sequences, failing to account for the heterogeneous information density across visual and textual tokens in LVLMs.

Main Contributions:
1. Introduces a calibration objective that uses attention aggregation for textual tokens and a contextual redundancy metric for visual tokens to reweight reconstruction loss.
2. Employs a two-scale quantizer that preserves high-magnitude outliers while maintaining resolution for non-outlier features.
3. Demonstrates consistent improvements across multiple LVLM backbones and benchmarks.

**Compliance With Llm Reviewing Policy:**

Affirmed.

**Final Justification:**

The authors have addressed most of my concerns; therefore, I will maintain my original score.

**Key Questions For Authors:**

The proposed method relies on a relatively small calibration set. The authors should provide a sensitivity analysis regarding both the size and the domain of the calibration data. Given the diverse downstream tasks of LVLMs, would performance degrade significantly if the calibration samples were reduced or drawn from a different distribution?

A key technical assumption in Section 3.2 is that outlier channels remain relatively stable within each layer. However, this claim lacks sufficient empirical support. Could the authors provide visualization or statistical evidence across different backbones to validate this? Specifically, I am concerned whether this assumption still holds for non-LLaVA architectures (e.g., encoder-decoder or fusion-heavy models).

**Limitations:**

yes

**Strengths And Weaknesses:**

Strengths:

- The paper identifies the modality gap in standard PTQ and provides a technically sound solution. By combining modality-specific importance metrics (attention for text, redundancy for vision) with an outlier-aware quantizer, it effectively addresses cross-modal error amplification.

- Evaluation across 7+ benchmarks and multiple configurations demonstrates consistent gains. Detailed ablation studies further confirm the individual contribution of each framework component.

- The manuscript is highly readable, featuring a logical structure and intuitive visualizations. The inclusion of Algorithm 1 and precise implementation details in Section 3.4 ensures a high level of reproducibility.

- ImpQuant advances PTQ by introducing fine-grained token-level modeling, moving beyond coarse-grained baselines. Its orthogonality to other efficiency techniques and its specific insights into object hallucination make it highly relevant for real-world LVLM deployment.

Weakness

- The reliance on a small calibration set raises concerns regarding data sensitivity across diverse downstream tasks. Furthermore, the core assumption that outlier channels remain stable across layers is stated without sufficient empirical validation across different model backbones.

- The evaluation is heavily centered on LLaVA-style (decoder-only) architectures. Testing on a broader range of LVLM design, such as encoder-decoder frameworks (e.g., Flamingo) or different fusion strategies, is necessary to demonstrate that the "heterogeneous density" assumption holds universally across the LVLM landscape.

---

> ### Author Rebuttal · Authors · 2026-03-30
>
> We sincerely thank the reviewer for the constructive and detailed feedback, and for recognizing the importance of addressing heterogeneous token importance in LVLM quantization, and the effectiveness of our fine-grained importance-aware calibration design. Below, we address each concern in detail with additional experiments and analysis.
>
> > **Q1: How sensitive is ImpQuant to the size and domain of the calibration set?**
>
> A1: ImpQuant is highly robust to the domain of the calibration set and stable regarding its size once a moderate threshold is reached. We conducted additional experiments on Qwen2.5-VL-7B under W3A16 by varying calibration set size. To assess domain sensitivity, we introduce LLaVA-Instruct as an out-of-distribution calibration source. Furthermore, to rule out sampling variance, we independently sampled 10 different 256-image subsets from MSCOCO and re-ran the full calibration and evaluation process for each group. Performance across different random subsets remains highly consistent (detailed in A5 to Reviewer rpM9). As shown in the table below, switching to LLaVA-Instruct causes negligible performance variation, confirming ImpQuant's robustness to domain shifts. Regarding the calibration set size, our token-importance metric requires a moderate amount of data to obtain reliable statistical estimates but does not benefit substantially from further scaling beyond 256. The selected size of 256 achieves a favorable trade-off between performance and computational cost.
>
> | Calibration Setting | Size | ScienceQA↑ | MMBench↑ | OCRBench↑ |MMMU↑
> |-|-|-|-|-|-|
> | MSCOCO | 64 | 85.07 | 80.68 | 80.40 | 45.57 |
> | MSCOCO | 128 | 85.94 | 81.01 | 81.60 | 46.24 |
> | MSCOCO | 200 | 86.23 | 81.25 | 81.80 | 46.44 |
> | MSCOCO (default) | 256 | 86.32 | 81.36 | 82.00 | 46.48 |
> | MSCOCO | 512 | 86.53 | 81.42 | 82.20 | 46.56 |
> | LLaVA-Instruct | 256 | 86.21 | 81.38 | 82.20 | 46.32 |
>
> > **Q2: Is the outlier channel stability assumption sufficiently validated?**
>
> A2: Thank you for the insightful comment. Our additional experiments empirically validate that the outlier channel is stable across inputs on varying backbones. To empirically validate this,  we randomly sampled 256 images from the MSCOCO dataset. For each layer in the LLM backbone, we extracted the indices of the top-1% outlier channels based on activation magnitudes for every single image. We then calculated the pairwise Jaccard similarity of these top-1% index sets across all 256 inputs to measure cross-input consistency. To verify architectural generalizability, we evaluated four distinct VLMs, specifically adding mPLUG-Owl3-7B to represent cross-attention-based fusion modules. Across all four backbones, the average Jaccard similarity of outlier channel indexes exceeds 0.94 and even for the layer with the lowest similarity, the value remains strictly above 0.92 across all models. This provides clear evidence supporting the observation that outlier channels remain highly stable across inputs.
>
> Model | Fusion Type | Avg Jaccard↑| Min Jaccard↑ | Max Jaccard↑
> |-|-|-|-|-|
> | LLaVA-OneVision-7B | MLP projector | 0.953 | 0.928 (Layer 4) | 0.989 (Layer 22) |
> | Qwen2.5-VL-7B | Dynamic ViT + LLM | 0.947 | 0.922 (Layer 2) | 0.986 (Layer 20) |
> | InternVL2-8B | InternViT+MLP | 0.958 | 0.931 (Layer 3) | 0.987 (Layer 24) |
> | mPLUG-Owl3-7B | Cross-attention | 0.941 | 0.920 (Layer 2) | 0.985 (Layer 22) |
>
> > **Q3: Generalizing to non-LLaVA architectures**
>
> A3: Our results confirm that ImpQuant generalizes well to non-LLaVA architectures. We add experiments on mPLUG-Owl3-7B, which employs a cross-attention-based fusion module, as a representative of fusion-heavy architectures—models that rely on dedicated cross-modal attention layers to integrate vision and language features, as opposed to simple concatenation-based designs. Our ImpQuant still outperforms baseline algorithms. To further verify the reliability of these gains, we performed paired t-tests. ImpQuant's improvements over the strongest baseline are statistically significant (p < 0.05) across all benchmarks. It shows that the "heterogeneous density" observation is a property of multimodal inputs, not of a specific architecture.
>
> |Bitwidth | Method | ScienceQA↑ | MMBench↑ | OCRBench↑ |MMMU↑|
> |-|-|-|-|-|-|
> | FP16 | - | 89.54 | 79.60 | 72.20 | 49.56 |
> || W3A16 | RTN | 85.90 | 76.22 | 65.20 | 45.51 |
> || AWQ | 87.25 | 77.42 | 67.80 | 46.78 |
> || MBQ | 86.82 | 76.83 | 67.60 | 46.42 |
> || **ImpQuant** | **87.86** | **77.45** | **70.80** | **48.12** |
> |W4A8 | RTN | 85.44 | 76.08 | 65.30 | 45.19 |
> || MBQ | 86.89 | 76.93 | 67.20 | 46.70 |
> || **ImpQuant** | **87.52** | **77.59** | **70.30** | **48.24** |
>
> We thank the reviewer again for the valuable feedback, which has strengthened our work. The additional experiments further confirm the generalizability and practical robustness of ImpQuant. We will incorporate all new results and discussions into the revised manuscript.

---

> > ### Author Rebuttal · Reviewer_Axuk · 2026-04-02
> >
> > My concerns have been mostly addressed. I keep my score.

---

> > > ### Author Response · Authors · 2026-04-02
> > >
> > > We sincerely thank you for your positive feedback. We are glad to learn that we have addressed your concerns and will incorporate your insightful suggestions and comments into our paper.

---

### Official Review · Reviewer_rpM9 · 2026-03-13

**Soundness:** 3
**Presentation:** 3
**Significance:** 3
**Originality:** 3
**Overall Recommendation:** 5
**Confidence:** 3

**Summary:**

This paper introduces ImpQuant, an importance-aware post-training quantization (PTQ) framework designed specifically for Large Vision-Language Models (LVLMs). It overcomes the limitations of previous methods that treat multimodal tokens homogeneously by explicitly reweighting the calibration loss to prioritize decision-critical visual and textual tokens. Furthermore, the framework incorporates a two-scale, outlier-aware activation quantization scheme to preserve essential high-magnitude channel features without degrading the resolution of normal channels. Extensive evaluations confirm that ImpQuant consistently enhances low-bit accuracy and minimizes object hallucinations across various architectures compared to state-of-the-art baselines

**Compliance With Llm Reviewing Policy:**

Affirmed.

**Key Questions For Authors:**

1. Could the authors provide the specific values used for the following hyperparameters across the different backbones: the balancing factor $\alpha$, the global scaling values $L_t$ and $L_v$, and the group size $G_l$ used for weight quantization?

2. How sensitive is the performance of ImpQuant to the choice of these hyperparameters ($\alpha, L_t, L_v$)? Have the authors conducted any sensitivity analysis to determine if the same values work across different LVLM architectures?

3. The importance weighting strategy is asymmetrical, using attention for text and redundancy for vision. Could the authors clarify the reasoning behind this choice? Specifically, was attention-based importance tested for visual tokens, or redundancy-based importance for text, and what were the outcomes?

4. In the ablation study (Table 4), the "outlier-aware activation quantization" seems to have a relatively minor impact on performance compared to the weight quantization improvements. Do the authors have further insights or experiments on specific scenarios where this component becomes more critical?

5. The visual redundancy probe requires $2N_v$ forward passes per image. Could the authors provide an estimate of the total time required for this offline calibration compared to standard PTQ? Additionally, have the authors analyzed how sensitive the importance weights are to the specific 256 images sampled? For example, do the results remain consistent if a different random subset of MSCOCO is used for calibration?

**Limitations:**

yes

**Strengths And Weaknesses:**

1. **Soundness**:


   - The proposed methodology appears technically sound and is supported by both empirical results and theoretical analysis. The authors provide a formal justification (Proposition 3.1) suggesting that highly redundant tokens induce near-zero curvature in the loss landscape, which would imply that quantization errors on these tokens result in a negligible penalty. The use of second-order error compensation in weight quantization follows established practices but is adapted via a novel token-importance matrix. The experimental evaluation is extensive, spanning diverse benchmarks and multiple LVLM architectures.
    - However, an analysis of the sensitivity of the proposed approach to key hyperparameters is not performed in the manuscript. Furthermore, the specific values used for several of these parameters (such as the number of groups $G_l$, the balancing factor $\alpha$, and the global scaling values $L_t, L_v$) do not appear to be explicitly indicated, which could potentially limit the transparency and reproducibility of the results.


---
2. **Presentation**:

    The submission is clearly written and well-structured, presenting the interplay between weight and activation quantization in a logical manner. The authors effectively distinguish their modality-specific strategies, and the narrative flow—from initial motivation to empirical validation—is easy to follow. The work is appropriately positioned within the context of current multimodal PTQ literature, and the methodological description is sufficiently detailed for the reader to understand the conceptual framework. However, the manuscript would benefit from a more thorough description of certain implementation details, such as specific hyperparameter values.

---
3. **Significance**:
   - This work addresses post-training quantization (PTQ), which is a highly relevant and timely topic. The motivation is well-presented, particularly the effort to disentangle the relative importance of visual versus textual tokens. This more nuanced treatment of multimodal sequences appears to provide genuine practical value; it is impressive that the quantized model matches or even occasionally outperforms the full-precision version on hallucination benchmarks (Table 3). Additionally, the framework achieves speedup numbers comparable to established baselines while maintaining higher accuracy.

    - However, while the proposed importance weights are interesting, the significance of the choice remains somewhat unclear. Specifically, the manuscript does not justify why attention-based importance was ruled out for visual tokens, nor why redundancy-based importance was not considered for text. A minor weakness in significance is also the seemingly limited impact of the "outlier-aware activation quantization" component, as suggested by the ablation results in Table 4.

---
4. **Originality**:

    - The authors provide an elegant solution by creatively adapting and extending known techniques for the LVLM context. For weight quantization, the injection of modality-aware importance weights into the Hessian-based error compensation is a novel and effective way to address the asymmetric roles of vision and text. Regarding activation quantization, while the problem of outliers has been addressed in prior work (e.g., SmoothQuant), the proposed solution of using two distinct scales for quantization is both simple and novel. Even though its final effectiveness remains somewhat unclear in the ablation study, it represents a fresh perspective on handling activation distributions in multimodal models.

---

---

> ### Author Rebuttal · Authors · 2026-03-30
>
> We thank the reviewer for recognizing the technical soundness of our methodology, the formal theoretical justification, the extensive experimental evaluation. We also appreciate the positive assessment of our originality. We address each concern below.
>
> > **Q1: Hyperparameter values and sensitivity analysis**
>
> A1: We apologize for the omission. The specific values used are: group size $|G_{l,g}|=128, \alpha=1.5, L_t=1.0, L_v=0.1$. The outlier channel ratio is fixed at 1% per layer. We conduct experiments on Qwen2.5-VL-7B under W4A8, sweeping each key hyperparameter while fixing the others. We visualize the plot depicting performance variations with hyperparameter changes at https://anonymous.4open.science/r/ImpQuant_supp_pic-C773. As shown in the figure, the performance of ImpQuant remains relatively stable when hyperparameters vary within a reasonable range.
>
> > **Q2: The choice of importance metrics.**
>
> A2: The asymmetric design is motivated by the fundamentally different information structures of the two modalities. Textual tokens are fewer in quantity and semantically dense. Aggregated attention naturally distinguishes content-bearing words (entities, attributes) from function words, providing a reliable and computationally cheap importance signal. In contrast, visual tokens are massively redundant. Attention scores for visual tokens tend to be diffuse and nearly uniform across patches especially in early-to-middle layers, making attention a poor discriminator of visual importance. Instead, our redundancy probe directly measures each patch's unique semantic contribution via representational dominance and leave-one-out ablation. To substantiate the design choice empirically, we will conduct an ablation comparing four different combinations of importance metrics on Qwen2.5-VL-7B (W4A8). The experimental results confirm that our asymmetric design optimally matches each metric to its modality's information structure.
>
> Text Metric | Vision Metric | ScienceQA↑| MMBench↑| OCRBench↑|MMMU↑
> |-|-|-|-|-|-|
> Attention|Redundancy (ImpQuant)| **86.02**|**79.74**|**82.90**|**46.04**
> Attention|Attention|85.61|79.28|81.80|45.85
> Redundancy | Redundancy | 85.19 | 78.86 | 79.70 | 45.47 |
> Redundancy|Attention|84.83|78.12|73.50|44.88
>
> > **Q3: Outlier-aware activation quantization shows modest gains.**
>
> A3: We address it from two complementary angles. First, outlier-aware activation quantization provides a cost-benefit tradeoff due to its negligible computational overhead. As shown in Table 5, ImpQuant incurs only <2% latency overhead compared to standard RTN quantization. The A-out component involves element-wise masking, separate scaling, and summation—all lightweight operations that do not alter the total MAC (Multiply-Accumulate Count). The memory overhead is also negligible: one binary mask and one more scalar scales per layer. Given this near-zero cost, even a modest accuracy improvement represents a highly favorable cost-benefit tradeoff.
>
> Second, the performance gains become substantial under more aggressive quantization settings. Under lower activation bitwidths, the outlier problem intensifies and the A-out component becomes critical. We validate this with W4A6 experiments.
>
> Bitwidth|Method|ScienceQA↑|MMBench↑|OCRBench↑|MMMU↑
> -|-|-|-|-|-
> W4A6|ImpQuant|**81.38**|**77.83**|**74.80**|**42.24**
> W4A6|w.o. A-out|80.71|77.15|73.60|41.67
>
> As shown above, dropping the A-out component under W4A6 leads to more significant performance degradation. Given that A-out incurs less than 2% latency overhead compared to standard RTN, even modest gains in higher bitwidths are favorable, while its contribution becomes essential for robust low-bit quantization.
>
> > **Q4: The time required for the calibration.**
>
> A4: We measured the time for offline calibration on a single NVIDIA H100 GPU with 256 MSCOCO images. Our method requires approximately 3 hours. The absolute time cost remains manageable for a one-time offline process.
>
> > **Q5: Sensitivity of sampled calibration data.**
>
> A5: We randomly sampled 10 different groups of 256-image subsets from MSCOCO and re-ran the entire calibration and evaluation process for each group on Qwen2.5-VL-7B (W3A16). We report the average and standard deviation of performance among the 10 samples. As shown below, the performance across different random subsets remains highly consistent indicated by a negligible standard deviation. This confirms that our method captures intrinsic model properties rather than overfitting to specific images. Besides, we evaluated the results under different calibration set sizes and domain shift. ImpQuant is robust to domain shift, as detailed in A1 to Reviewer Axuk.
>
> | Model | Bitwidth|ScienceQA↑|MMBench↑|OCRBench↑|MMMU↑
> |-|-|-|-|-|-|
> |Qwen2.5-VL-7B|W3A16 | 86.31 $\pm$ 0.009 | 81.37$\pm$ 0.012 | 82.02$\pm$ 0.015 | 46.49 $\pm$ 0.011
>
> We thank the reviewer for the constructive and insightful feedback. We will incorporate these results and analyses in the later versions.

---

> > ### Author Rebuttal · Reviewer_rpM9 · 2026-04-01
> >
> > I thank the authors for their thorough rebuttal, which has addressed all my concerns. I will increase my score accordingly. In the final manuscript, please integrate the detailed discussion and alternative configurations regarding the text-vision importance weight asymmetry. Additionally, include the results for aggressive quantization settings to properly substantiate the empirical gains of the outlier-aware activation quantization.

---

> > > ### Author Response · Authors · 2026-04-02
> > >
> > > We sincerely thank you for the positive feedback and for acknowledging that our rebuttal has adequately addressed the concerns. We also appreciate the helpful and insightful suggestions and will incorporate them into the final manuscript.

---

### Official Review · Reviewer_Wm2J · 2026-03-16

**Soundness:** 1
**Presentation:** 3
**Significance:** 2
**Originality:** 2
**Overall Recommendation:** 3
**Confidence:** 3

**Summary:**

This paper introduces ImpQuant, a Post-Training Quantization (PTQ) framework for Large Vision-Language Models (LVLMs). The authors observe that standard, modality-agnostic PTQ methods fail to account for the heterogeneous information density in multimodal inputs. To address this, ImpQuant reweights the quantization calibration objective using fine-grained, token-level importance scores. These scores are derived from aggregated attention for text tokens and a dual-probe redundancy metric for visual tokens. Additionally, the paper proposes a channel outlier-aware activation quantization scheme that uses a two-scale quantizer to preserve extreme activation values without collapsing the resolution of normal channels. The authors demonstrate consistent performance improvements over standard PTQ baselines (like RTN, AWQ, and SmoothQuant) across multiple LVLM backbones and diverse VQA benchmarks.

**Compliance With Llm Reviewing Policy:**

Affirmed.

**Final Justification:**

The rebuttal partially resolved my concerns and I raise my score accordingly.

**Key Questions For Authors:**

Given the concerns raised in the weaknesses section, could you clarify the rationale for using the proposed importance metrics for text and visual tokens?

**Limitations:**

Yes.

**Strengths And Weaknesses:**

**Strenght**

1. High Relevance: Tackles the critical and timely problem of multimodal quantization and heterogeneous information density in LVLMs.

2. Clear Presentation: Well-written with intuitive motivations and effective visual aids (e.g., Figures 1 & 2).

3. Comprehensive Evaluation: Thorough benchmarking across three diverse modern LVLM backbones and a wide array of VQA and hallucination tasks.

**Weaknesses**

1. Unjustified Modality Discrepancy in Importance Estimation.

The paper proposes completely disjointed methodologies for estimating token importance across modalities: aggregated attention for text and a dual-probe heuristic for vision. Given that both text and visual tokens are projected into the same LLM embedding space and processed by the same Transformer attention layers, the necessity of abandoning the attention-based metric for visual tokens is poorly justified.

2. Positional Bias in Textual Importance Formulation

The aggregated attention metric for textual tokens (Equation 2) lacks positional normalization. Because the LVLM utilizes an auto-regressive causal mask, earlier tokens naturally receive attention from a much larger pool of subsequent tokens compared to later tokens. By employing a simple sum over all $k$, the formulation mathematically guarantees a heavy bias toward early sequence tokens. It is highly likely that this metric is simply acting as a proxy to protect early structural tokens, rather than genuinely isolating semantic density as claimed. This feels so obviously wrong that I have to say please let me know if my understanding of the process is incorrect.

3. Methodological Flaws in Visual Redundancy Probes

The two heuristics proposed to measure visual token redundancy are questionable under the mechanics of modern Vision-Language Models:
- Out-of-Distribution (OOD) Degradation: Probe 1 measures confidence by tiling a single patch across all spatial positions. This destroys the 2D spatial structure and positional embeddings the model relies on, pushing the input out of the training distribution. Consequently, the resulting "confidence" score is likely an artifact of the model breaking down rather than a reliable measure of semantic dominance.
- Blindness to Spatial Correlation: Probe 2 utilizes a leave-one-out ablation (JSD) to measure contextual necessity. In high-resolution images, large objects are distributed across many neighboring patches. Masking a single patch will almost certainly yield a near-zero change in the prediction distribution due to the high redundancy of neighboring patches.

4. Lack of Rigorous Ablations for Metric Design

The empirical validation of the proposed metrics is incomplete. While Table 4 ablates the overall presence of the importance module, there are no isolated ablations to justify the specific design of the heuristics themselves. The authors do not evaluate Probe 1 versus Probe 2 independently, nor do they compare the dual-probe method against a baseline that simply uses attention aggregation for vision. Furthermore, the paper lacks statistical analysis or density plots of the intermediate values (e.g., the JSD scores or confidence values). Without verifying that these metrics actually yield a well-distributed, meaningful signal, the heuristics remain unvalidated.

---

> ### Author Rebuttal · Authors · 2026-03-30
>
> We sincerely thank the reviewer for the constructive feedback. We address each concern with both analytical clarifications and experimental evidence.
>
> > **Q1: Modality Discrepancy in Importance Estimation.**
>
> A1: Please refer to A2 to Reviewer rpM9, where we provide ablations and statistical analyses supporting this design.
>
> > **Q2: Positional Bias in Textual Importance Formulation**
>
> A2: We address this concern from both analytical and experimental perspectives. Analytically, we clarify that our metric captures global structural necessity and local semantic density, both of which are crucial for preserving LVLM performance under quantization. Empirically, we conduct ablation experiments that disentangle positional bias from semantic density.
>
> First, recent literatures [1,2] demonstrate that initial tokens often act as "attention sinks". They are structurally indispensable for stabilizing the attention manifold, even if their semantic content is low. Quantization errors on these early tokens disproportionately collapse the model's generation capabilities. Therefore, the mathematical bias towards early tokens in Equation 2 is actually a desirable property that implicitly protects the structural integrity of the generation process.
>
> Second, despite the decaying trend, the metric captures true semantic density locally. Text token importance in Eq. (2) is primarily determined by attention from generated output tokens, directly reflecting each token's influence on the final answer. Within the sequence, content-bearing words (e.g., visual attributes) exhibit sharp attention spikes compared to adjacent function words, successfully biasing the quantization budget towards semantic peaks.
>
> To empirically disentangle positional bias from semantic density, we conduct ablation studies on textual token weighting strategies using Qwen2.5-VL-7B under W3A16. We compare:
>
> 1. Uniform calibration across all text tokens.
> 2. High importance (5$\times$) applied to the first 20% of the textual tokens, with uniform weights for the rest. Tests whether simply protecting early tokens is effective.
> 3. Position-Normalized Attention: Aggregated attention normalized by the number of legally attending tokens, removing causal positional bias.
> 4. ImpQuant.
>
> As shown below, increasing the weights of early tokens leads to performance improvement, demonstrating that early tokens are structurally indispensable for stabilizing the attention manifold. ImpQuant formulation also demonstrates advantages compared to Normalized Attention.
>
> Textual Importance|ScienceQA↑|MMBench↑|OCRBench↑|MMMU↑
> -|-|-|-|-
> Uniform|85.31|79.98|75.70|45.91
> Emphasize Early-Tokens|85.62|80.24|78.90|46.03
> Normalized Attn|85.98|80.97|81.60|46.26
> ImpQuant|**86.32**|**81.36**|**82.00**|**46.48**
>
> > **Q3: Methodology in Visual Redundancy Probes**
>
> A3: Thank you for the insightful comment. ImpQuant has already taken into account the two mentioned scenarios. We discuss each probe in turn.
>
> Probe 1: Due to massive pre-training, LVLMs are highly robust and do not simply collapse into artifacts from single-token tiling. Probe 1 acts as an information-density check to isolate a token's absolute semantic strength. If a token contains vague background, the top-1 output probability is low. If it contains distinct, critical information, the top-1 probability is high. This reliably measures a token's independent importance.
>
> Probe 2: The phenomenon you described, masking a patch yielding near-zero JSD due to neighboring redundancy, is exactly what we aim to capture. A minimal distribution change confirms the token is highly redundant, justifying a lower importance weight $p_i$. This is rigorously supported by Proposition 3.1 (Eqs. 8-9). We also prove theoretically that high spatial correlation corresponds to small residual energy $\xi_i$, rendering the loss landscape locally flat regarding that token's quantization error. Both our theory and experiments validate JSD as an effective redundancy measure.
>
> > **Q4: Additional Ablations for Metric Design**
>
> A4: To address the incomplete ablation study, we conduct fine-grained ablations isolating each probe and comparing against alternative metrics-attention aggregation for vision on Qwen2.5-VL-7B(W3A16). As shown below, by combining probe 1 and 2, ImpQuant achieves better performance than using either probe alone or employing aggregated attention. We visualize the density plots of JSD scores at https://anonymous.4open.science/r/ImpQuant_supp-F1FD.
>
> Visual Importance|ScienceQA↑|MMBench↑|OCRBench↑|MMMU↑
> -|-|-|-|-
> Uniform|85.31|79.98|75.70|45.91
> Probe 1 Only|84.83|78.58|77.20|45.74
> Probe 2 Only|85.57|80.62|80.90|46.09
> Aggregated attention|85.73|80.44|80.60|46.14
> Probe 1+Probe 2 (ImpQuant)|**86.32**|**81.36**|**82.00**|**46.48**
>
> [1] Xiao, G. Efficient streaming language models with attention sinks (ICLR 2024)
>
> [2] Kang, Seil, et al. See what you are told: Visual attention sink in large multimodal models. (ICLR 2025)

---

### Decision · Program_Chairs · 2026-04-30

**Decision:**

Accept (regular)

**Comment:**

This paper proposes an importance-aware post-training quantization framework for large vision-language models, addressing heterogeneous token importance across modalities. The problem is timely and relevant, and the paper is clearly written with solid empirical evaluation across multiple backbones and benchmarks.

The main concerns raised by reviewers focus on the asymmetric design of importance metrics, potential positional bias in textual attention, and the validity of the visual redundancy probes. In the rebuttal, the authors provide additional ablations and clarifications. The positional bias concern is reasonably addressed both analytically and empirically, and the added ablations help support the effectiveness of the proposed importance metrics. However, the justification for the specific design choices remains somewhat empirical, and some questions about robustness are only partially resolved.

It also appears that one reviewer may not have fully taken the rebuttal into account. Considering the full discussion, I find that the major concerns are sufficiently addressed, though not completely eliminated.

Overall, I lean toward accept. The paper provides a meaningful and practical contribution, and the remaining issues are primarily about depth of justification rather than fundamental flaws.